# Materialized Knowledge Bases from Commonsense Transformers

**Tuan-Phong Nguyen**
Max Planck Institute for Informatics
Saarland Informatics Campus
Saarbrücken, Germany
`tuanphong@mpi-inf.mpg.de`

**Simon Razniewski**
Max Planck Institute for Informatics
Saarland Informatics Campus
Saarbrücken, Germany
`srazniew@mpi-inf.mpg.de`

## Abstract

Starting from the COMET methodology by Bosselut et al. (2019), generating commonsense knowledge from commonsense transformers has recently received significant attention. Surprisingly, up to now no materialized resource of commonsense knowledge generated this way is publicly available. This paper fills this gap, and uses the materialized resources to perform a detailed analysis of the potential of this approach in terms of precision and recall. Furthermore, we identify common problem cases, and outline use cases enabled by materialized resources. We posit that the availability of these resources is important for the advancement of the field, as it enables an off-the-shelf-use of the resulting knowledge, as well as further analyses on its strengths and weaknesses.

## 1 Introduction

Compiling comprehensive collections of commonsense knowledge (CSK) is an old dream of AI. Besides attempts at manual compilation (Liu and Singh, 2004; Lenat, 1995; Sap et al., 2018) and text extraction (Schubert, 2002; Tandon et al., 2014; Mishra et al., 2017; Romero et al., 2019; Nguyen et al., 2021a), commonsense knowledge compilation from pretrained language models (Bosselut et al., 2019; Hwang et al., 2021; West et al., 2021) has recently emerged. In 2019, Bosselut et al. introduced *Commonsense Transformers* (COMET), an approach for fine-tuning language models on existing corpora of commonsense assertions. These models have shown promising performance in generating commonsense assertions after being trained on established human-authored commonsense resources such as ATOMIC (Sap et al., 2018) and ATOMIC$_{20}^{20}$ (Hwang et al., 2021).

More recently, West et al. (2021) extracts commonsense assertions from a general language model, GPT-3 (Brown et al., 2020), using simple prompting techniques. Surprisingly, using this machine-authored commonsense corpus to fine-tune COMET helps it outperform GPT-3, which is 100x larger in size, in terms of commonsense capabilities.

Despite the prominence of this approach (the seminal COMET paper (Bosselut et al., 2019) receiving over 300 citations in just two years), to date, no resource containing commonsense knowledge compiled from any COMET model is publicly available. As compilation of such a resource is a non-trivial endeavour, this is a major impediment to research that aims to understand the potentials of the approach, or intends to employ its outputs in downstream tasks.

This resource paper fills this gap. We fine-tune the COMET pipeline on two established resources of concept-centric CSK assertions, CONCEPTNET (Speer et al., 2017) and ASCENT++ (Nguyen et al., 2021a), and execute the pipeline for 10K prominent subjects. Unlike the ATOMIC resources, which were used to train COMET in (Bosselut et al., 2019; Hwang et al., 2021) and have their main focus on events and social interactions, the two resources of choice are mostly about general concepts (e.g., *lions can roar*, or *a car has four wheels*). Furthermore, as those two resources were constructed using two fundamentally different methods, crowdsourcing and web text extraction, it enables us to discover potentially different impacts they have on the COMET models.

By taking the top-10 inferences for each subject-predicate pair, we obtain four resources, CONCEPTNET (GPT2-XL, BART) and ASCENT++ (GPT2-XL, BART), containing 900K to 1.4M ranked assertions of CSK. We perform a detailed evaluation of the intrinsic quality, including fine-grained precision (typicality and saliency) and recall of each resource, derive qualitative insights into the strengths and weaknesses of the approach, and highlight extrinsic use cases enabled by the resources.

Our contributions are:

1. The materialization of the COMET approach for two language models (GPT2-XL, BART) on two concept-centered commonsense knowledge bases (CONCEPTNET, AS-CENT++);

2. Quantitative and qualitative evaluations of the resulting resources in terms of precision, recall and error categories, showing that in terms of recall, COMET models outperform crowd-sourced construction and are competitive with web text extraction, while exhibiting moderate gaps in terms of precision to both;

3. Illustrative use cases of the materialized resources in statement aggregation, join queries, and search.

The materialized resources, as well as an interactive browsing interface, are available at `https://ascentpp.mpi-inf.mpg.de/comet`.

## 2 Related work

Early approaches at CSK compilation relied on expert knowledge engineers (Lenat, 1995) or crowd-sourcing (Liu and Singh, 2004), and the latter approach has recently been revived (Sap et al., 2018). To overcome scalability limitations of manual compilation, text extraction is a second popular paradigm. Following early attempts on linguistic corpora (Mishra et al., 2017), increasingly approaches have targeted larger text corpora like Wikipedia, book scans, or web documents (Tandon et al., 2014; Romero et al., 2019; Nguyen et al., 2021a,b), to build CSK resources of wide coverage and quality.

Recently, both approaches have been complemented by knowledge extraction from pre-trained language models: Language models like BERT (Devlin et al., 2019) or GPT (Radford et al., 2019; Brown et al., 2020) have seen millions of documents, and latently store associations among terms. While West et al. (2021) used prompting to extract symbolic CSK from GPT-3, Bosselut et al. (2019) proposed to tap this knowledge by supervised learning: The language models are fine-tuned on statements from existing knowledge resources, e.g., trained to predict the object *Africa* when given the subject-predicate pair *elephant, At-Location*, based on the ConceptNet triple ⟨*elephant, AtLocation, Africa*⟩. After training, they can be used to predict objects for unseen subject-predicate pairs, e.g., locations of wombats.

The approach gained significant attention, and variants are employed in a range of downstream tasks, e.g., commonsense question answering (Bosselut and Choi, 2020), commonsense explanation (Wang et al., 2020), story generation (Guan et al., 2020), or video captioning (Fang et al., 2020).

Yet, to date, no materialized knowledge resource produced by any COMET model is available (AUTOTOMIC from (West et al., 2021) being based on prompting GPT-3). The closest to this is a web interface hosted by the AllenAI institute at `https://mosaickg.apps.allenai.org/model_comet2020_entities`. However, this visualizes only predictions for a single subject, making, e.g., aggregations or count impossible, and only shows top-5 predictions, and without scores.

## 3 Methodology

We follow the implementations in the official code repository[1] of the COMET-ATOMIC$_{20}^{20}$ project (Hwang et al., 2021) to compute assertions, and decide on output thresholds.

**Training CSKBs.** We use two established concept-centered commonsense knowledge bases (CSKBs), CONCEPTNET 5.7 (Speer et al., 2017) and ASCENT++ (Nguyen et al., 2021a) as training resources, considering 13 CSK predicates from each of them: *AtLocation*, *CapableOf*, *Causes*, *Desires*, *HasA*, *HasPrerequisite*, *HasProperty*, *HasSubevent*, *MadeOf*, *MotivatedByGoal*, *PartOf*, *UsedFor* and *ReceivesAction*.

1. CONCEPTNET (Speer et al., 2017) is arguably the most widely used CSKB, built by crowd-sourcing. CONCEPTNET 5.7 is its lastest version[2], consisting of 21 million multilingual assertions, spanning CSK as well as general linguistic and taxonomic knowledge. We retain English assertions only, resulting in 207,210 training assertions for the above-mentioned predicates.

2. ASCENT++ (Nguyen et al., 2021a) is a project aiming for automated CSK extraction from large-scaled web contents based on open information extraction (OpenIE) and judicious

[1]`https://github.com/allenai/comet-atomic-2020/`
[2]`https://github.com/commonsense/conceptnet5/wiki/Downloads`

| Parameter | GPT2-XL | BART |
|---|---|---|
| num_beams | 10 | 10 |
| temperature | 1.0 | 1.0 |
| top_p | 0.9 | 1.0 |
| repetition_penalty | 1.0 | 1.0 |
| max_length | 16 | 24 |
| no_repeat_ngram_size | 0 | 3 |
| early_stopping | True | True |
| do_sample | False | False |

Table 1: Configurations for beam-search decoders.

cleaning and ranking approaches. The AS-CENT++ KB consists of 2 million English CSK assertions for the 13 mentioned predicates.

**Language models.** We consider two autoregressive language models (LMs) that were also used in the original COMET paper, GPT2-XL (Radford et al., 2019) and BART (Lewis et al., 2020).

**Materialization process.** We query the fine-tuned COMET models for 10,926 subjects in CONCEPTNET which have at least two assertions for the 13 CSK predicates. For each subject-predicate pair, we use beam search to obtain completions, with different configurations (see Table 1) for BART and GPT2-XL, following the parameters specified in the published code repository and models. We retain the top-10 completions for each subject-predicate pair, with their *beam scores* (i.e., sum of log softmax of all generated tokens) returned by the *generate* function[3] of the Transformers library (Wolf et al., 2020).

**Output.** The resulting resources, CONCEPTNET (GPT2-XL, BART) and ASCENT++ (GPT2-XL, BART), contain a total of 976,296 and 1,420,380 and 1,271,295 and 1,420,380 assertions after deduplication, respectively, as well as their corresponding beam scores. All are available for browsing, as well as for download, at `https://ascentpp.mpi-inf.mpg.de/comet` (see screenshot of browsing interface in Figure 2).

## 4 Analysis

We perform three kind of analyses: (1) a quantitative evaluation of the intrinsic quality of the assertions, based on crowdsourcing, (2) a qualitative

---

[3] `https://huggingface.co/docs/transformers/main/en/main_classes/text_generation#transformers.generation_utils.GenerationMixin.generate`

evaluation that outlines major strengths and weaknesses, and (3) an illustration of use cases enabled by both resources.

### 4.1 Quantitative evaluation

The original paper (Bosselut et al., 2019) only evaluated the top-1 triple per subject-predicate pair. Furthermore, it solely evaluated triples by plausibility, which is a necessary, but only partly a sufficient criterion for being considered commonsense (Chalier et al., 2020).

In the following, we evaluate samples from the generated resources along two *precision* dimensions, typicality (top-100 assertions per subject) and saliency (top-10 assertions per subject). We also evaluate *recall*, by measuring the degree to which each resource covers the statements in a human-generated ground truth.

**Precision: Typicality and saliency.** Following Romero et al. (2019); Nguyen et al. (2021a), we assess assertions in the CSK resources along two precision dimensions: *typicality* and *saliency*, which measure the degree of truth and the degree of relevance of assertions, respectively. We use the Amazon Mechanical Turk (AMT) platform to obtain human judgements. Each dimension is evaluated based on a 4-point Likert scale and an option for *no judgement* if the annotator is not familiar with the concepts. Assertions are transformed into human-readable sentences using the templates introduced by Hwang et al. (2021). Each assignment is done by three different workers. Following Hwang et al. (2021), any CSK assertion that receives the two higher scores in the Likert scale is labelled as *Typical* or *Salient*, and the two lower scores as *Untypical* or *Unsalient*. The final judgements is based on majority vote.

In terms of sampling process, for typicality, we draw 500 assertions from each resource when restricting to top-100 assertions per subject. For saliency, we pick 500 random samples from the pool of top-10 assertions per subject.

Results are reported in the left part of Table 2. We see a significant drop in the quality of assertions in the LM-based generations compared to the training resources. In terms of the neural models, for both training CSKBs, the BART models demonstrate better typicality than the GPT2-XL ones. Assertions in BART-ASCENT++ also have significantly better saliency than in GPT2-XL-ASCENT++. Interestingly, BART-CONCEPTNET

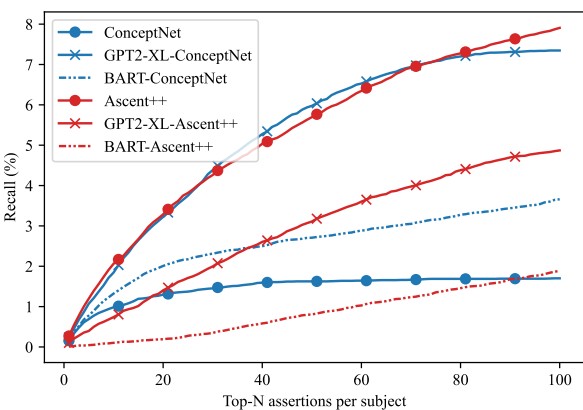

Figure 1: Resource recall in relation to resource size, at similarity threshold $t = 0.98$.

is nearly on par with ASCENT++ on both metrics.

**Recall.** We reuse the CSLB dataset (Devereux et al., 2014) that was processed by Nguyen et al. (2021a) as ground truth for recall evaluation. The CSLB dataset consists of 22.6K human-written sentences about property norms of 638 concepts. To account for minor reformulations, following Nguyen et al. (2021a), we also use embedding-based similarity to match ground-truth sentences with statements in the CSK resources. We specifically rely on precomputed SentenceTransformers embeddings (Reimers and Gurevych, 2019). We also restrict all CSK resources to top-100 assertions per subject.

The evaluation results are shown in the right part of Table 2, where we report recall at similarity thresholds 0.96, 0.98 and 1.0, as well as resource size. We also plot the recall values at different top-N assertions per subject in Figure 1 with similarity threshold $t = 0.98$. As one can see, ASCENT++ outperforms both COMET models trained on it even though it is significantly smaller. We see opposite results with the CONCEPTNET-based resources, where the COMET models generate resources of better coverage than its training data. Our presumption is that the LMs profits more from manually curated resources like CONCEPTNET, but hardly add values to resources that were extracted from the web, as LMs have not seen fundamentally different text. Furthermore, in contrast to precision, GPT2-XL models have better results than BART models in terms of recall, on both input CSKBs.

## 4.2 Qualitative observations

LMs have the strength to generate an open-ended set of objects, even for subjects seen rarely or not at all in the training data. For example, while CONCEPTNET stores only one location for *rabbit*: *"a meadow"*, both BART- and GPT2-XL-CONCEPTNET can generalize to other correct locations, such as *wilderness, zoo, cage, pet store*, etc. In the recall evaluation, we pointed out that CONCEPTNET, a manually-built CSK resource with relatively small size, considerably benefits from LMs generations as they improve the coverage of the resource substantially.

However, as indicated in the precision evaluation, LM generations are generally of lower precision than those in the training data. Common error categories we observe are:

- **Co-occurrence misreadings:** LMs frequently predict values that merely frequently co-occur, e.g., ⟨*locomotive, atLocation, bus stop*⟩, ⟨*running, capableOf, put on shoes*⟩, ⟨*war, desires, kill people*⟩, ⟨*supermarket, capableOf, buy milk*⟩.

- **Subject-object-copying**: LMs too often repeat the given subject in predictions. For instance, 45 of 130 objects generated by BART-CONCEPTNET for the subject *chicken* also contain *chicken*, such as ⟨*chicken, CapableOf, kill/eat/cook chicken*⟩ or ⟨*chicken, UsedFor, feed chicken*⟩.

- **Quantity confusion**: LMs struggle to distinguish quantities. For example, GPT2-XL-CONCEPTNET generates that *bike* has *four wheels* (top-1 prediction), and then also *two wheels* (rank 3), *three wheels* (rank 4) and *twelve wheels* (rank 5). The weakness of dealing with numbers is known as a common issue of embeddings-based approaches (Berg-Kirkpatrick and Spokoyny, 2020).

- **Redundancy**: Generated objects often overlap, bloating the output with redundancies. Most common are repetitions of singular/plural nouns, e.g., the top-2 generations by BART-CONCEPTNET for *doctor-CapableOf*: *"visit patient"* and *"visit patients"*. Redundancies also include paraphrases, e.g., ⟨*doctor, CapableOf, visit patients / see patients*⟩; or ⟨*doctor, CapableOf, prescribe medication / prescribe drug / prescribe medicine*⟩ (GPT2-XL-ASCENT++ generations). Clustering might alleviate this issue (Nguyen et al., 2021a).

| Resource | Typicality@100 | | Saliency@10 | | Recall@100 | | | Size@100 |
|---|---|---|---|---|---|---|---|---|
| | **Typical** | **Untypical** | **Salient** | **Unsalient** | **t=0.96** | **t=0.98** | **t=1.00** | **#triples** |
| Ascent++ | **78.4** | **11.0** | **62.8** | **34.6** | **8.9** | **7.9** | **4.6** | 202,026 |
| GPT2-XL-Ascent++ | 57.2 | 27.4 | 37.2 | 58.4 | 6.0 | 4.9 | 2.6 | 1,091,662 |
| BART-Ascent++ | 69.8 | 17.4 | 50.6 | 42.6 | 2.6 | 1.9 | 1.0 | 1,092,600 |
| ConceptNet | **93.6** | **3.6** | **80.0** | **16.8** | 2.3 | 1.7 | 0.9 | 164,291 |
| GPT2-XL-ConceptNet | 66.6 | 21.4 | 63.8 | 32.6 | **9.0** | **7.3** | **3.8** | 967,343 |
| BART-ConceptNet | 72.6 | 17.0 | 63.4 | 33.4 | 5.3 | 3.7 | 1.0 | 1,092,600 |

Table 2: Intrinsic evaluation (Typicality, Saliency and Recall - %) and size of CSK resources.

## 4.3 Downstream use of materialized resources

Beyond systematic evaluation, materialized resources enable a wide set of downstream use cases, for example context-enriched zero-shot question answering (Petroni et al., 2020), or KB-based commonsense explanation (Wang et al., 2020). We exemplarily illustrate four enabled types of basic analyses, (1) frequency aggregation, (2) join queries, (3) ranking and (4) text search.

**Frequency aggregation.** Materialized resources enable to count frequencies. In Table 3, we demonstrate the three most common objects for each predicate in the GPT2-XL-ConceptNet resource. Interestingly, the third most common location of items in the KB is *"sock drawer"*, which is only ranked as the 190th most common location in ConceptNet. Similarly, the top-3 objects for *CapableOf* in the generated KB rarely occur the training data.

**Join queries.** One level further, materialized knowledge enables the construction of join queries. For example, we can formulate conjunctive queries like:

- Animals that eat themselves include *chicken*, *flies*, *grasshopper*, *mice*, *penguin*, *worm*.

- The most frequent subevents of subevents are: *breathe*, *swallow*, *hold breath*, *think*, *smile*.

- The most common parts of locations are: *beaches*, *seeds*, *lot of trees*, *peel*, *more than one meaning*.

**Ranking.** Since statements in our materialized resources come with scores, it becomes possible to locally and globally rank assertions, or to compare statements pairwise. For example, in GPT2-XL-ConceptNet, the triple ⟨*librarian, AtLocation, library*⟩, which is at rank 140, has a score

| Predicate | Most common objects |
|---|---|
| AtLocation | desk (3210), cabinet (2481), sock drawer (1771) |
| CapableOf | branch out (963), branch off (747), taste good (556) |
| Causes | death (2504), tears (1290), happiness (1254) |
| Desires | eat (949), have fun (816), sex (742) |
| HasA | more than one meaning (1387), seeds (1316), peel (1170) |
| HasPrerequisite | metal (1965), plastic (1594), water (1423) |
| HasProperty | good (2615), useful (2585), good for (1746) |
| HasSubevent | breathe (1006), swallow (721), take off shoes (658) |
| MadeOf | plastic (1427), aluminum (1297), wood (905) |
| MotivatedByGoal | have fun (994), enjoyment (493), succeed (444) |
| PartOf | new testament (914), human experience (683), alabama (667) |
| ReceivesAction | found in house (1110), eaten (800), found in hospital (779) |
| UsedFor | cooking (627), decoration (454), transport (448) |

Table 3: Most common objects generated by GPT2-XL-ConceptNet. Numbers in parentheses indicate frequency of the corresponding objects.

of −0.048, which is much higher than that of ⟨*elephant, CapableOf, climb tree*⟩ (score = −0.839, ranked 638,048 globally).

**Text search.** Finally, we can use materialized resources for text search. For example, we can search in GPT2-XL-ConceptNet for all assertions that include the term *"airplane"*, finding expected matches like ⟨*airplane, AtLocation, airport*⟩ and ⟨*flight attendant, CapableOf, travel on airplane*⟩, as well as surprising ones like ⟨*scrap paper, UsedFor, making paper airplane*⟩ and ⟨*traveling, HasSubevent, sleeping on airplane*⟩.

## 5 Conclusion

We introduced four CSKBs computed using two COMET models (BART and GPT2-XL) trained on two existing CSK resources (ConceptNet and Ascent++). Our findings are:

1. The COMET methodology produces better results on modest manually curated resources (CONCEPTNET) than on larger web-extracted resources (ASCENT++).

2. COMET's recall can significantly outperform that of modest manually curated ones (CONCEPTNET), and reach that of large web-extracted ones (ASCENT++).

3. In terms of precision, a significant gap remains to manual curation, both in typicality and saliency. To web extraction, a moderate gap remains in terms of statement typicality.

We also identified common problems of the COMET generations, such as co-occurrence misreadings, subject copying, and redundancies, which may be subject of further research regarding post-filtering and clustering.

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

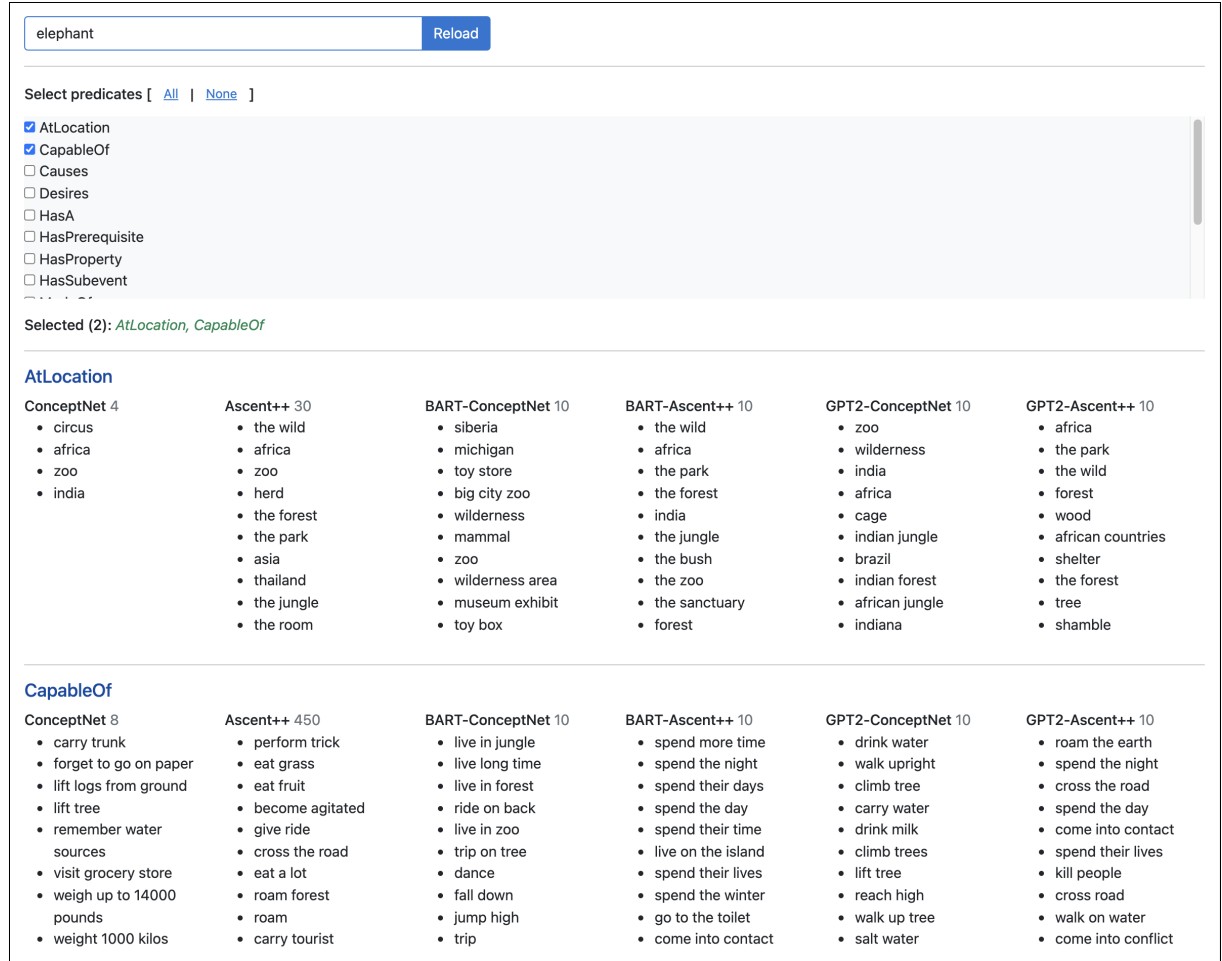

Figure 2: Web interface showing top-10 assertions per predicate in six CSK resources. The number in grey next to a CSKB indicates the total number of assertions for the corresponding subject-predicate pair in the KB.

Niket Tandon, Gerard de Melo, Fabian M. Suchanek, and Gerhard Weikum. 2014. WebChild: harvesting and organizing commonsense knowledge from the web. In *WSDM*.

Cunxiang Wang, Shuailong Liang, Yili Jin, Yilong Wang, Xiaodan Zhu, and Yue Zhang. 2020. Semeval-2020 task 4: Commonsense validation and explanation. In *SemEval*.

Peter West, Chandra Bhagavatula, Jack Hessel, Jena D Hwang, Liwei Jiang, Ronan Le Bras, Ximing Lu, Sean Welleck, and Yejin Choi. 2021. Symbolic knowledge distillation: from general language models to commonsense models. *arXiv preprint arXiv:2110.07178*.

Thomas Wolf, Lysandre Debut, Victor Sanh, Julien Chaumond, Clement Delangue, Anthony Moi, Pierric Cistac, Tim Rault, Remi Louf, Morgan Funtowicz, Joe Davison, Sam Shleifer, Patrick von Platen, Clara Ma, Yacine Jernite, Julien Plu, Canwen Xu, Teven Le Scao, Sylvain Gugger, Mariama Drame, Quentin Lhoest, and Alexander Rush. 2020. Transformers: State-of-the-art natural language processing. In *EMLNP: System Demonstrations*.
