# OpenReview forum: "Materialized Knowledge Bases from Commonsense Transformers"
_aclweb.org/ACL/2022/Workshop/CSRR — ACL 2022 Workshop CSRR_

### Official Review · Reviewer_Yejc · 2022-03-22
**Trained COMET models analyzed for precision+recall and used to generate new commonsense KGs -- but not clear why we need this resource?**

**Rating:** 5
**Confidence:** 4

**Review:**

What the paper is about: The authors offer a new resource generated from COMET models trained on commonsense knowledge graphs like ConceptNet and Ascent++. They study not just plausibility but also the precision (typicality and saliency) as well as recall of the models' predictions. They analyze different base LMs and datasets on these metrics and offer insights. Finally, they demonstrate a web interface with wider customizations than the original one hosted by AI2.

Key shortcoming: The authors call it a resource paper (L043). However, the benefit of the new "generated" commonsense knowledge graphs is not well established. Section 4.3 hints at some use cases like aggregation, joins, ranking, and text search. But the benefit of having a static set of predictions (this new resource) is not clear. (1) How are these better than the base KGs like ConceptNet and Ascent++? Perhaps they are bigger but not always more salient/typical/exhaustive than the original KGs (see Table 2). (2) How are these better than retaining the trained COMET model, which can generate such inferences and many more, on demand?

Pros: good analysis + useful resource.
Cons: usefulness of the resource not demonstrated.

EDIT: Reviewer jbiB rightly points to a major missing related work, which further challenges the paper's claim to novelty.

Minor:
- Should you be referring to Untypical as Atypical instead? I was fairly confident that the latter is "correct" but words have no inherent meaning anyway so this is up to the authors.
- Saliency vs Typicality could benefit from a formal definition (in English, not just in a formula) each. Do they differ in just values of k for top k extractions?

---

### Official Review · Reviewer_8czw · 2022-03-23
**Strong paper that fills a gap in current commonsense knowledge extraction work**

**Rating:** 9
**Confidence:** 5

**Review:**

This paper proposes to materialize neural commonsense predictions with COMET into concrete resources. The paper investigates two SotA knowledge bases (ConceptNet and ASCENT++) and two standard language models (GPT-2-XL and BART). The evaluation estimates the precision of the generated knowledge through salience and typicality, and the recall by comparison against a feature norms dataset, CSLB. The results indicate the promise of this approach, but also point to key obstacles in terms of redundancy, subject copying, and co-occurrence misreading.

The paper is overall well-written, original, and the evaluation is solid. The pointed challenges are thought-provoking.

The paper size is in between a short and a long paper, so it is unclear to me whether this paper qualifies as long of short. If this is meant to be a long paper, it would be good to include more discussion on how would the authors propose to circumvent the key challenges with this knowledge base generation method. These mitigation strategies are currently only briefly listed in the conclusion, which leaves many questions unanswered. Furthermore, some quantitative investigation of how would the downstream applications benefit from the created sources despite these challenges would be useful.

---

### Official Review · Reviewer_jbiB · 2022-03-24
**Interesting and original work but the proposed method might need further development before the resulting CSKB can be used as a reliable resource**

**Rating:** 5
**Confidence:** 3

**Review:**

This paper studies generating commonsense knowledge directly from pre-trained language models for two CSKBs - ConceptNet and Ascent++. The direction is interesting and original, and the paper is well written and easy to follow. However,

1. The paper claims that "up to now no materialized resource of commonsense knowledge generated via pre-trained language models is publicly available.". However it's not true. West et al. (2021) construct AUTOTOMIC via GPT3 that's 10x larger than ATOMIC, and provide comprehensive and in-depth analysis and evaluation.
2.  Lack of novelty: the proposed method directly applies previous COMET pipeline on two established CSKBs without further improvement or adaption.
3. Evaluation shows a clear gap between proposed PLM generated CSKs and original human written CSKs. Without further filtering or purification, it's questionable whether the generated noisy CSKB can be used as a reliable resource.

Citation:
West, Peter et al. “Symbolic Knowledge Distillation: from General Language Models to Commonsense Models.” ArXiv abs/2110.07178 (2021): n. pag.

---

### Decision · Program_Chairs · 2022-03-28

Accept